# Multi-scale ResNet and BiGRU automatic sleep staging based on attention mechanism

Changyuan Liu[1]*, Yunfu Yin[1], Yuhan Sun[1], Okan K. Ersoy[2]

**1** School of Measurement and Communication Engineering, Harbin University of Science and Technology, Harbin, China, **2** Electrical and Computer Engineering Department, Purdue University, West Lafayette, Indiana, United States of America

* liuchangyuan@hrbust.edu.cn

**Data Availability Statement:** All relevant data are within the manuscript and its Supporting information files.

**Funding:** This project is funded by the National Natural Science Foundation of China (51779050)

## Abstract

Sleep staging is the basis of sleep evaluation and a key step in the diagnosis of sleep-related diseases. Despite being useful, the existing sleep staging methods have several disadvantages, such as relying on artificial feature extraction, failing to recognize temporal sequence patterns in the long-term associated data, and reaching the accuracy upper limit of sleep staging. Hence, this paper proposes an automatic Electroencephalogram (EEG) sleep signal staging model, which based on Multi-scale Attention Residual Nets (MAResnet) and Bidirectional Gated Recurrent Unit (BiGRU). The proposed model is based on the residual neural network in deep learning. Compared with the traditional residual learning module, the proposed model additionally uses the improved channel and spatial feature attention units and convolution kernels of different sizes in parallel at the same position. Thus, multi-scale feature extraction of the EEG sleep signals and residual learning of the neural networks is performed to avoid network degradation. Finally, BiGRU is used to determine the dependence between the sleep stages and to realize the automatic learning of sleep data staging features and sleep cycle extraction. According to the experiment, the classification accuracy and kappa coefficient of the proposed method on sleep-EDF data set are 84.24% and 0.78, which are respectively 0.24% and 0.21 higher than the traditional residual net. At the same time, this paper also verified the proposed method on UCD and SHHS data sets, and the figure of classification accuracy is 79.34% and 81.6%, respectively. Compared to related existing studies, the recognition accuracy is significantly improved, which validates the effectiveness and generalization performance of the proposed method.

## 1 Introduction

More than a third of a person's day is spent in sleep, sleep plays a vital role in the balance of physiological function. Sleep staging is the basis of sleep quality assessment, and the accuracy and convenience of sleep staging are the key factors in the diagnosis of sleep-related diseases [1]. At present, the medical analysis of sleep state is mainly by reading the information of Poly-SomnGram (PSG) for sleep stage interpretation. During the whole night sleep, various signals of the human body will show different characteristics with the change of sleep state. According

and the National Natural Science Foundation of Heilongjiang Province (F2016022). All the funds obtained for this study were used to purchase corresponding experimental equipment and conduct experimental tests, and the researcher did not receive salary from the sponsors. The funders had no role in study design, data collection and analysis, decision to publish, or preparation of the manuscript.

**Competing interests:** No author has a competing interest.

to the change of signal characteristics, the sleep process can be divided into several sleep stages, corresponding to the change of sleep state. In the General Rechtschaffen & Kales (R&K) Guidelines [2], all night's sleep is mainly divided into Wake stage (W), Rapid Eye Movement stage (REM) and S1, S2, S3 and S4. The American Academy of Sleep Medicine suggests that both S3 and S4 are in deep sleep, which merged S3 and S4 into Slow wave Sleep stage (SS) [3].

At present, there are two sleep stages in the field of sleep medicine. One is the classification of manual sleep periods by scholars, which takes hours to observe and analyze. The other is automatic sleep staging, which is the current mainstream research method. In the initial stage of sleep staging research, EEG can only be divided into various sleep stages by experts. This work is not only time-consuming and cumbersome, but also prone to subjective errors [4]. Such studies have employed algorithms such as decision trees [5–9], support vector machines [10–13], Markov models [14, 15], and neural networks [16, 17], which operate on combinations of the traditional multi-channel PSG biometrics to provide algorithmic and automated assessment of a patient's underlying sleep architecture. How to solve the tedious and time-consuming problems of manual calibration method and effectively improve the accuracy of sleep staging has become a research hotspot of experts around the world.

With the development of deep learning, several studies have attempted to develop an automatic sleep staging model based on deep learning. A convolutional neural network framework based on joint classification and prediction is proposed in literature [18], which can automatically extract sleep features from raw data, but the experimental results are still not as good as high-level classification models based on feature engineering at the same time. Then the framework of recurrent neural network is improved, and the end-to-end hierarchical recurrent neural network model for automatic sleep staging of sequence pairs is established, and good staging results are achieved [19, 20], However, there are limitations on the sleep data with small-scale imbalanced categories, and the classification accuracy has great room for improvement. In order to improve the effect of sleep staging, Jia Ziyu et al. used parallel convolution network to automatically learn the original EEG features, and then used empty convolution and residual link to fuse features [21]. Later, some scholars applied Recurrent Neural Networks (RNNs) in biological signal processing, and achieved good results. Supratak et al. [22] applied RNNs on sleep staging research, which has good effect but can improve the space greatly. However, RNNs has defects in processing long sequences, so LSTM came into being. Luo et al. [23] applied LSTM to sleep stages recognition of EEG signals. Kuo et al. [24] proposed an automatic sleep stage scoring combining the techniques of data augmentation, ensemble convolutional neural network (CNN), and expert knowledge. In addition, residual-like fusion structure is used to append the attention map to the input feature map for adaptive feature refinement [25]. The accuracy of sleep staging has been greatly improved. Feng et al. [26] proposed an automatic sleep staging algorithm based on the time attention mechanism. This approach reduces computing resources and time costs. In 2021, Altini Marco et al. [27] comprehensively analyzed the influence of accelerometer, peripheral signal mediated by autonomic nervous system (ANS) and circadian rhythm characteristics on the detection of sleep stage of large data set, making the accuracy of sleep stage reached 79%. Its limitations are however ecological validity is limited by wearing reference PSG, which can be a disruption in the participant's typical sleep patterns. In 2021, Huttunen et al. [28] used CNNs to stage EEG signals. But the model had a weak ability to learn the relationship between sleep fragments, with a recognition rate of 68.7%. Casciola et al. [29] built a CNN-LSTM network model for sleep staging, lacking the access to important information and multiscale extraction of features. In 2022, You Yuyang et al. [30] came up with a novel method for automatic sleep stage classification based on the time, frequency, and fractional Fourier transform (FRFT) domain features extracted from a single-channel electroencephalogram (EEG). Although this method achieves

a certain effect on sleep staging of EEG signals on sleep-EDF data set, the classification accuracy (81.6%) is still 2.64% lower than the method proposed in this paper. The drawbacks of this method are that it only extracts the features of a single channel and does not fully extract the sleep features of multiple channels.

Considering the above-given factors, this paper provides a detailed review of the residual network, which is proven to be effective in sleep staging. As a result, we propose an EEG-based sleep signal staging model using Multi-scale Attention Residual Nets (MAResnet) and a Bidirectional Gated Recurrent Unit (BiGRU) network. This study have proposed three innovations as follows:

Firstly, in order to highlight the characteristics of EEG sleep sequence, channel feature attention unit and spatial feature attention unit are added to the residual module, while the ReLU activation function is replaced by the extended exponential linear unit activation function to construct the residual spatial channel attention module. This plays a key role in identifying different sleep periods, since it can fully learn the importance of different channel characteristics and the correlation between characteristics.

Secondly, convolution kernels of different sizes were used in parallel at the same spatial position. The shortcomings of the single-size filter extraction were mitigated, which allowed us to obtain the multi-scale EEG sleep feature output and fully excavate the intrinsic sleep information of the EEG signal.

Thirdly, given the limited learning capability of the sleep cycle characteristics caused due to the traditional algorithm's inability of identifying timing patterns in the long-term correlation data, a BiGRU network was introduced to analyze the timing information in detail.

The improved model has stronger learning ability for features, which can fully excavate more intrinsic features, identify the temporal pattern of long-term association data, and effectively improve the accuracy of sleep staging model. Experimental results show that the improved model has better classification performance.

## 2 Theoretical analysis

### 2.1 Residual neural net

The convolutional neural network has a wide range of applications in the field of image classification and target recognition, from 5 layers of the convolution of original Lenet to 19 layers of convolution of VGG -19. The recognition effect and depth of this network are closely related. Networks such as VGGNet and GoogLeNet both indicate that having enough depth is a prerequisite for the model to perform well, but when the network depth increases to a certain extent, deeper networks mean higher training errors. The reason for the increase of error is that the deeper the net, the more prone to gradient dispersion. Generally speaking, the depth of convolutional network is easy to appear the gradient attenuation problem of shallow hidden layer, which increases the difficulty of training. In this paper, the residual network is selected as the basic network to improve. Residual network can effectively alleviate the problem of gradient attenuation.

The principle of Residual Networks (ResNet) is as follows. If the $x$ is input to a neural net, it is expected to output as function $H(x)$. From the diagram, the desired output of the underlying mapping is to determine the residual function fitting a given layer, $H(x)$ is the desired underlying mapping description. To ensure that the depth increases while the network does not degenerate, y = x layers need to be superimposed on the shallow net. The input x is transmitted and output in the corresponding ' jump connection ' mode, and the corresponding result is $H(x) = F(x) + x$. Under $F(x) = 0$, this relation is simplified to $H(x) = x$. The relation is the

specific analysis shows that in jump connection mode, computational complexity and additional parameters do not increase.

This is equivalent to ResNet changing the learning goal, corresponding to the difference between $H(x)$ and $x$. In the subsequent training process, the goal to be achieved is to reduce the residual error as much as possible to make it zero, which shows that the accuracy of the output results does not decrease after the network depth increases. For an optimized identity map, the comparison and analysis show that the processing difficulty of setting the residual difference "0" is obviously lower than that of fitting the identity map through complex nonlinear layers.

After introducing "jump connection", the gradient returns to the shallow layer through "1", which can avoid gradient dispersion and effectively deal with the network degradation problem after depth increase. This does not introduce additional parameters, so learning to find identity maps significantly reduces the difficulty. The following formula is the expression corresponding to the structure of the residual learning block in Eq (1).

$$F(x) = W_2 \sigma(W_1 x) \tag{1}$$

where $\sigma$ is a nonlinear function ReLU, connecting to the next ReLU by shortcut and outputting y after calculation.

$$Y = F(x, \{W_i\}) + x \tag{2}$$

When the dimension of input and output changes, X linear transformation is implemented based on "jump connection", and $W_s$ is processed to match the dimensions. The expression is as follows:

$$Y = F(x, \{W_i\}) + W_s x \tag{3}$$

The number of layers of the residual block should be higher than two, and the residual block can not meet the lifting requirements when the residual block is one layer. The residual structure can be simply written as follows.

$$x_{l+1} = x_l + F(x_l, W_l) \tag{4}$$

In the formula, $x_L$ is the characteristic of any deep element $L$ and $x_l$ is the characteristic of shallow element $l$. Through recursion, when the depth of element $L$ is any value, its characteristic $x_L$ can be obtained by the sum of the characteristic $x_l$ of the shallow element $l$ and the residual function $\sum_{i=l}^{L-1} F$.

$$x_L = x_l + \sum_{i=l}^{L-1} F(x_i, W_i) \tag{5}$$

## 2.2 Bi-directional gated loop unit network

Sleep data contain a lot of time sequence information, so we can use cyclic neural network to learn the time information in EEG signal and give the result of sleep cycle judgment. The cyclic neural network is improved to adapt each cycle unit to the dependence of different time scales, and the gate recurrent unit network (GRU) is obtained. Fig 1 shows the GRU structure. The

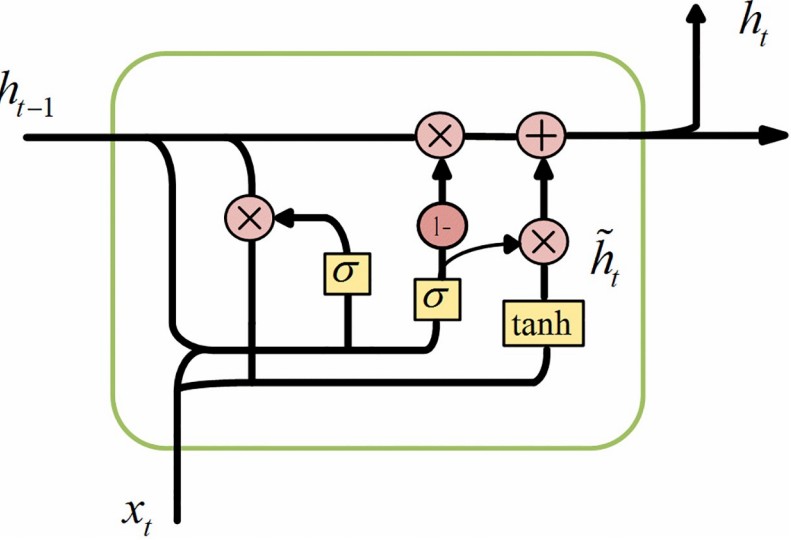

**Fig 1. GRU structure.**

parameters can be updated by Eq (6).

$$
\begin{aligned}
r_t &= \sigma(W_r x_t + U_r h_{t-1}) \\
z_t &= \sigma(W_z x_t + U_z h_{t-1}) \\
\tilde{h}_t &= \tanh(W x_t + U(r_t, h_{t-1})) \\
h_t &= (1 - z_t) h_{t-1} + z_t, \tilde{h}_t
\end{aligned}
\tag{6}
$$

In the formula, $W_r$, $W_z$, $W$, $U_r$, $U_z$, and $U$ respectively represent the weight matrix of GRU, $\sigma$ is a logical sigmoid function, and $z_t$ represents the update gate, which can determine the update degree of the GRU unit's activation value. The $r_t$ represents the reset gate, whose update process is similar to that of $z_t$. The $\tilde{h}_t$ represents the candidate hidden layer and $h_t$ represents the hidden layer.

However, GRU is a one-way neural network algorithm, in which all state transmissions are carried out in one direction. In the process of equipment failure prediction, we introduce BiGRU, considering that the output of the current time is related to the state of the previous and subsequent moments. The BiGRU neural network model is determined by the state of two GRUs, which are unidirectional in the opposite direction. This means that for each time point in a given sequence, BiGRU has complete sequence information about all the time points before and after it. The parameters can be updated by Eq (7).

$$
\begin{aligned}
z_t &= \vec{h}_t \oplus \overleftarrow{h}_t \\
\vec{h}_t &= \mathrm{GRU}\left(x_t, \vec{h}_{t-1}\right) \\
\overleftarrow{h}_t &= \mathrm{GRU}\left(x_t, \overleftarrow{h}_{t+1}\right)
\end{aligned}
\tag{7}
$$

It can be seen that the hidden layer state of BiGRU in time is determined by the current

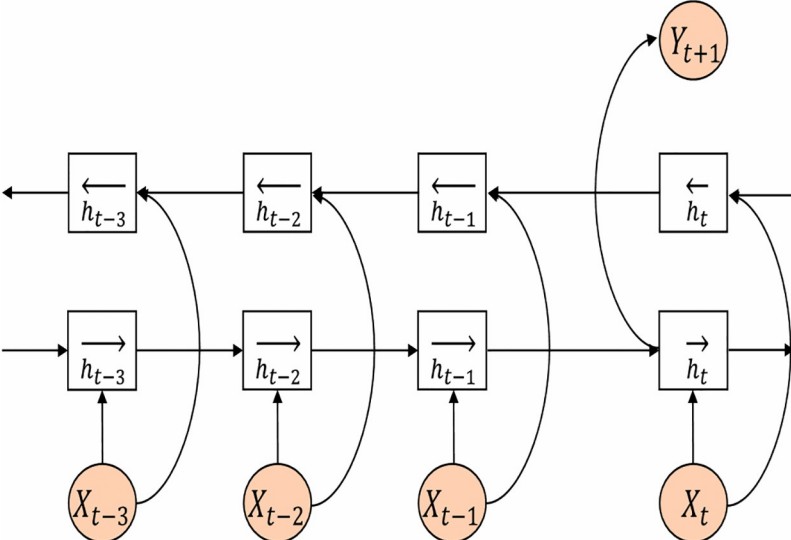

**Fig 2. Two-way GRU structure.**

input $x_t$, forward hidden layer state and backward hidden layer state, where $\oplus$ indicates the operation to connect two vectors. In this paper, BiGRU is introduced into the network when processing EEG time series data. The model structure of BiGRU is shown in Fig 2.

In Fig 2, X1, X2, Xt are time series data, the output is divided into two states, $\overrightarrow{h_t}$, $\leftarrow h_t$ are positive and reverse hidden states. BiGRU can process sleep information from two directions through two completely independent GRUs, which can better mine information in two-way time structure.

## 3 Attention mechanisms

### 3.1 Model principles

The attention model is essentially a set of weight coefficients independently learned through the network, which emphasizes the region of interest in a "dynamic weighting" manner while inhibiting the mechanism of unrelated background regions. The correlation degree between different features and the important information of each channel is not the same, which needs to reflect the different characteristics of different channels and the degree of relevance of important information. Hence, this paper introduces the attention model into sleep recognition research and thus constructs a channel attention unit (CAU). The CAU is for the direct processing of the information in a channel, which may ignore the information interaction in space. To this end, a spatial attention unit (SAU) is constructed. The SAU is for the equal treating of features in each channel, which may ignore the information interactions between the channels. Therefore, the CAU and SAU are added to the original residual module in this paper to construct the residual attention module, which plays a key role in identifying different sleep states.

### 3.2 CAU mechanism

Essentially, the internal mechanism of the channel feature attention unit represents a degree of similarity. The closer the input to the target is, the higher the weight is, and the more relevant

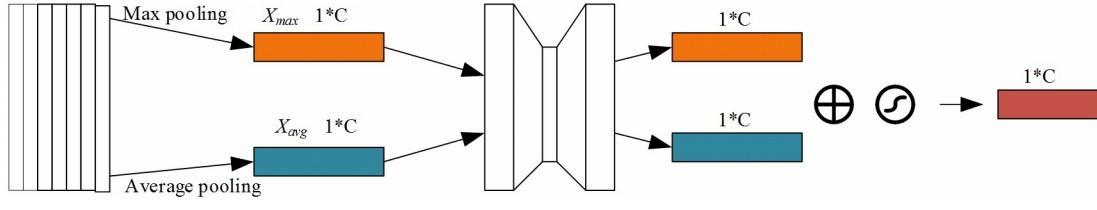

**Fig 3. CAU mechanisms.**

the state of the input is to the output. In this paper, the maximum and average pooling methods are used to process the input information. Thus, the feature vectors, *Xavg* and Xmax*max*, are obtained after maximum and average pooling. As shown in Fig 3, the CAU first integrates the input data according to the direction of the time sequence. Then, the Multilayer Perceptron (MLP) calculates Xavg*avg* and Xmax*max*, and obtains two new feature vectors. Here, the required channel feature weight vectors can be obtained by summing the elements one by one. Finally, the channel feature weight vector is multiplied by the input feature vector to obtain the input feature required by the spatial attention unit. The number of neurons in the input and output layers of the MLP neural network corresponds to the number of input characteristic channels. That is mainly used to construct the contribution degree of *Xavg* and *Xmax* input channel characteristic information.

CAU formula is shown in Eq (8).

$$
\begin{aligned}
M(X) \quad &= d(MLP(Avgpool(X)) + MLP(MaxPool(X))) \\
&= d(W_1(W_0(X_{avg})) + W_1(W_0(X_{max})))
\end{aligned}
\tag{8}
$$

where $W_0 \in R^{C/r \times C}$ and $W_1 \in R^{C/r \times C}$ are the weights between the input layer and the output layer with the hidden layer, and $X_{avg} \in R^{1 \times C}$ and $X_{max} \in R^{1 \times C}$ are the eigenvectors obtained after representing the average pooling and the maximum pooling, $\delta$ is Sigmoid activation functions, and the output is limited between 0 and 1.

### 3.3 SAU mechanisms

The spatial feature attention unit takes the feature map output by the channel attention module as the input feature map of this module. Firstly, a maximum pooling and average pooling of the feature graph based on the channel attention module output are performed to obtain the feature vectors *Lmax* and *Lavg*. Then, *Lmax* and *Lavg* are merged, after which convolution feature fusion is performed. Then, the feature weight vector is obtained by Sigmoid function and

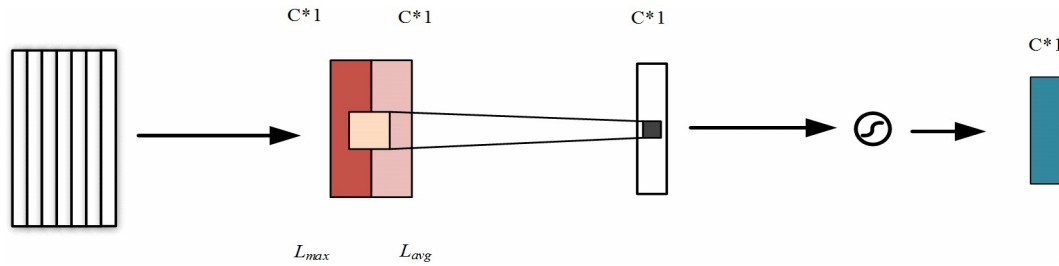

**Fig 4. SAU mechanisms.**

multiplied by the input feature vector to obtain the final attention feature. The underlying mechanism of the SAU is shown in Fig 4.

SAU formula is shown in Eq (9).

$$M(L) = s(f^{7\times7}([AvgPool(L); MaxPool(L)]))$$
$$= s\left(f^{7\times7}\left(\left[L_{avg}; L_{\max}\right]\right)\right) \tag{9}$$

where, $\delta$ is the Sigmoid activation function, the output is between 0 and 1, $X_{avg} \in R^{1\times C}$ and $X_{\max} \in R^{1\times C}$ represents the vector obtained after average pooling and maximum pooling.

## 3.4 Improved residual attention module

Traditional ResNets do not learn the key features, i.e., they neither learn the relevance and importance of different features in each input channel nor the relevance and importance of different channel features. To address this issue, this paper improves the traditional residual module. The CAU and SAU introduced in Sections 3.2 and 3.3 are added to the traditional residual module so that they can learn the correlation relationship and the degree characteristics of different channels, which play a key role in identifying different sleep states. To standardize the data batch and speed up the training, Batch Normalization layers are added after each convolutional layer in the residual module. The Relu activation function having the advantage of unilateral inhibition is adopted, and a new module called residual space channel attention module (RSCAM) is created, as shown in Fig 5.

## 3.5 Build MAResnet-BiGRU network model

In the original residual net, the convolutional kernels are all 1×3 of size, and they struggle with analyzing data comprehensively due to the scale constraints. If a single-size filter is selected when extracting the features of the EEG sleep signal, the results will be quite limited. Hence, it is difficult to obtain more input features, and the convolutional kernels cannot meet the input diversity requirements. However, if multiple kernels of different sizes are used in the same module layer during feature extraction, the robustness to the changes in the EEG sleep signal can be enhanced. Thus, this paper improves the original residual neural network and establishes a MAResnet, where the EEG signal can pass through, to obtain the specific characteristics of the timing signal and the information carried. Since the characteristics obtained in this process are in the form of time series, it is necessary to make a detailed and comprehensive analysis of them by establishing a model. Fig 6 illustrates the schematic diagram of the proposed MAResnet-BiGRU model.

To improve the expression ability of the EEG signal in the convolution layer, the convolution kernel should be used to extract the sleep characteristics of the EEG signal at different scales. In the improved network model, the improved residual attention module proposed in Section 3.4 is added to the residual net. The first layer uses a standard convolution layer with a convolution kernel size of 1×7. After the convolution calculation of the first layer, the maximum pool processing of 1×3 is performed. Next, the convolution kernels of sizes 1×3, 1×5, and 1×7 are used in parallel in the residual attention module. In the neural network at each scale, the number of convolution kernels in the residual attention module is set to 64, 128, 256, and 512. Each convolution kernel corresponds to the scale of output, sharing the same convolution kernel parameters and so the convolution of a scale. They check the EEG signals' processing characteristics, which form the electrical characteristics of the three scale output types. Hence, by achieving the goal of multi-scale study, they can make convolution layer "observation" sleep EEG signals from multiple scales, so that the inherent characteristics of sleep can

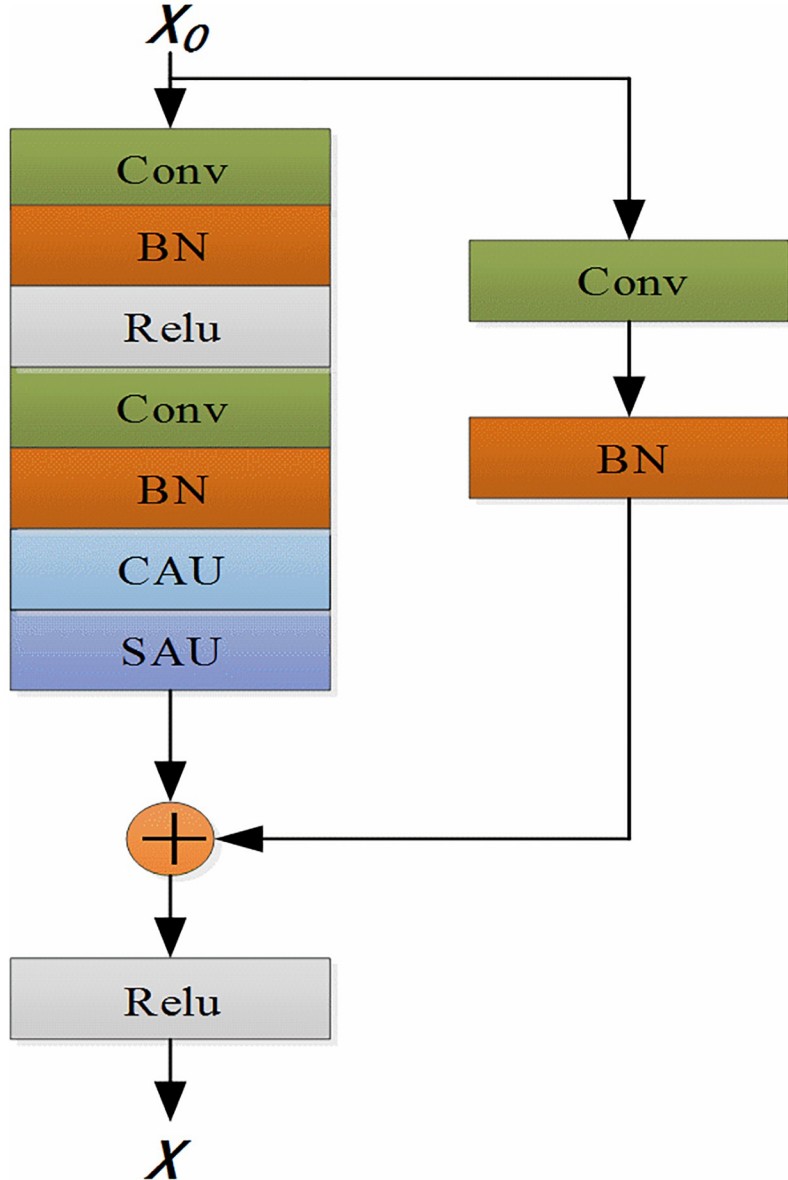

**Fig 5. RSCAM overall module.**

more fully tap EEG signals. One of the important factors in determining the classification effect is the depth and parameter quantity of the neural network model. However, this study does not use local max and average pooling operations to reduce the calculation of parameters and avoid overfitting. Instead, the global average, the local maximum, and the average pooling operations make each feature map get a corresponding value. Thus, the output dimension of the neural network at each scale is 1×512.

After improving the residual net, we can obtain the specific characteristics of the timing signal and the information carried. Since the characteristics obtained in this process are in the form of time series, we need to perform a comprehensive analysis of them by building a model. The BiGRU introduced in Subsection 2.2 is added to the network model and the *tanh*

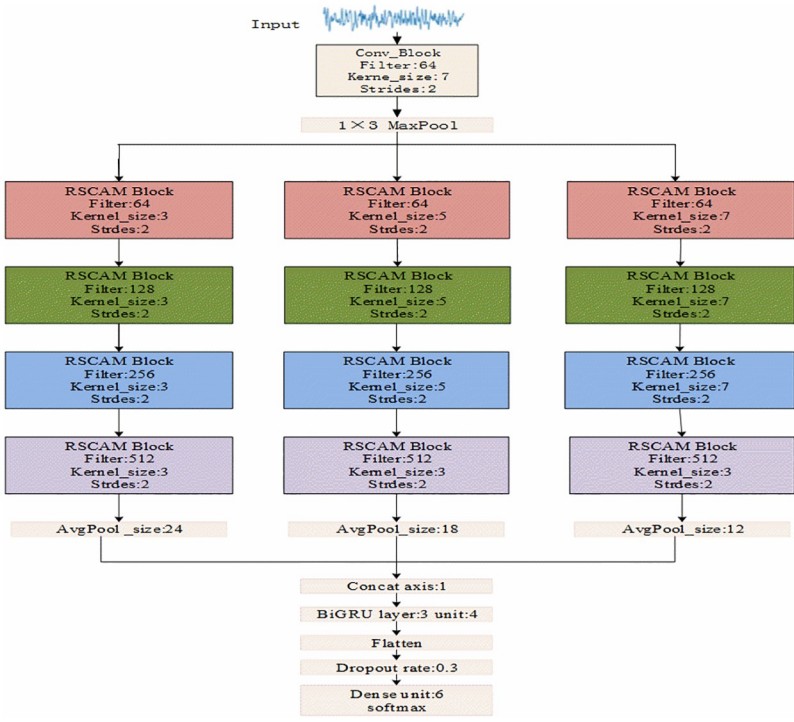

**Fig 6. Overall structure of the model.**

function is selected as its activation function. Finally, the fusion feature is used as the input of the full connection layer, and the dropout mechanism is added after that. This can effectively avoid overfitting occurring due to memory training set samples, which negatively affect the network performance. The regularization method of the dropout is to discard the nodes with a random selection probability of p $p$ in a certain layer of the network and a certain iteration during the training process, and then continue discarding process in the next iteration. That helps to obtain a model with good generalization ability until the end of the training.

## 4 Experimental design and verification

### 4.1 Data sources and preprocessing

To evaluate the performance of MAResnet-BiGRU, three datasets are used in the experiments-Sleep-EDF data set, UCD data set and SHHS data sets.

Sleep-EDF data set [31, 32]. The data set contains polysomnography of 20 healthy subjects PSG, each recorded approximately 20 h long. All EEG signals have the same sampling frequency of 100 Hz, and the signals are divided into 30 s each segment. According to R&K guidelines, the experts manually labeled the data as W, S1, S2, S3, S4 and REM stages. This paper combined the S3 stage and N4 stage of R&K sleep standard into the same stage as the S3 stage according to the standard of sleep. Table 1 lists the classification results of sleep experts on Sleep-EDF dataset. According to the staging of the Sleep-EDF dataset, the sample category of the experimental dataset is unbalanced. The proportion of samples in S2 sleep stage and S1 sleep stage is about 6:1. The number of samples in S2 sleep stage accounts for 42.43% of the total sample, and the remaining four sleep stages account for 57.57% of the total sample.

**Table 1. Sleep -EDF sleep staging.**

| index | quantity | Proportion |
|:---:|:---:|:---:|
| W | 7927 | 18.90 |
| S1 | 2804 | 6.68 |
| S2 | 17799 | 42.43 |
| S3 | 5703 | 13.59 |
| REM | 7717 | 18.40 |

UCD data set [31]. Provided by St Vincent University Hospital / Dublin University College sleep apnea database, The data set is composed of 25 patients' overnight PSGs, and the sampling rate includes 128 Hz. After removing the records containing unmarked data, 20 overnight PSGs are finally selected.

SHHS data set [33, 34]. The data set is from the National Heart Lung & Blood Institute which is aiming to study sleep disordered breathing. 5445 records are involved. Each record has 14 polysomnography (PSG) channels and the signal acquisition frequency is 125.0 Hz. In this experiment, C3 / A2 and C4 / A1 EEG channels are adopted.

In experiments, there are usually a small number of error labels in the sample, which will affect the effect of prediction. At this point, we introduce label smoothing to solve this problem. In the sample, if some labels are wrong, then at the time of training, the sample may have a negative impact on the training results. But if we have a way to "tell" the model that the label of the sample is not necessarily correct, then the model will be trained to exclude a small number of "bad samples" self-identification. In the multi-class training experiment, the positive sample in the label category of a given data set is set to 1 and the negative sample is set to 0, and the vector is shown in Eq (10).

$$P_i = \begin{cases} 1, & \text{if } (i = y) \\ 0, & \text{if } (i \neq y) \end{cases} \tag{10}$$

The real category of y as the target and the i is one of the multiple categories. This leads the model to believe too much in the predicted category, and when the data is incorrectly labeled, the training results are biased. At the same time, when the distribution of data is uneven, too much dependence on a large number of labels will lead to overfitting. Therefore, the label smoothing method is used to change the real probability distribution of labels to Eq (11):

$$P_i = \begin{cases} 1 - \varepsilon, & \text{if } (i = y) \\ \dfrac{\varepsilon}{k - 1}, & \text{if } (i \neq y) \end{cases} \tag{11}$$

The k is the total classification category, the $\varepsilon$ is the error rate, and the probability of 1-$\varepsilon$ of the new label after randomization is the same as that of the original label.

In network training, the optimal prediction probability distribution is as follows:

$$Z_i = \begin{cases} \log \dfrac{(k - 1)(1 - \varepsilon)}{\varepsilon} + \alpha, & \text{if } (i = y) \\ \alpha, & \text{if } (i \neq y) \end{cases} \tag{12}$$

Where $\alpha$ can be any real number. Therefore, in the training of network model, the addition of label smoothing can suppress the phenomenon of overfitting to a certain extent, increase the generalization ability of the network in this paper, and improve the classification effect.

## 4.2 Assessment/ classification experiments

The experimental environment of this paper is: Pytorch open source deep learning framework in python3 environment under Inter (R) Core (TM) i5-3317U-CPU-GTX1080Ti, 64bit, Windows 10 system. To verify the feasibility of the MAResnet-BiGRU net, the model proposed in this paper was applied to the Sleep-EDF data set, UCD data set and SHHS data set to study and analyze five sleep stages.

An optimization algorithm is required for minimizing the loss function and helping the network model to produce the optimal results. The Adaptive moment estimation(Adam) method returns the adaptive learning rate by using different network parameters that are more stable. Hence, the Adam optimizer, with parameters $\beta 1 = 0.9$, $\beta 2 = 0.999$ and epsilon = 1e-8, is selected. Set the initial learning rate to 0.005, the weight decay ratio to 0.1, and the batch-size to 16. 200 epochs per training. Using the same parameters, the proposed MAResnet-BiGRU model and the other four network models are applied to the Sleep-EDF dataset for comparative analyses. The other four network models are: i) the single-scale ResNet before improvement, ii) the MAResnet after improvement, ii) the improved single-scale ResNet with BiGRU (ResNet-BiGRU), and iv) the CNN-BiGRU. The classification results were evaluated based on the recall rate and the classification recognition rate of the total sleep cycle. The classification training in the open-source framework PyTorch resulted in the above-given five network models. Table 2 shows the results.

As can be seen from Table 2, under the label smoothing treatment, the sleep staging recognition rate of the single-scale traditional ResNet model was lowest, only 62.20%, where the S1 stage recognition rate was even lower, only 49.32%. This can be explained by three main reasons. First, all the convolution cores in the traditional ResNet are 1×3 of size in the operation process, so the convolution layer cannot "observe" data from multiple scales. Second, although the traditional ResNet can solve the degradation problem in the deep network to a certain extent, it does not learn the relevance and importance of the features of different channels of input. Third, the network cannot recognize the timing pattern in a long time associated data, which results in a low recognition rate. The sleep cycle classification recognition rate of the ResNet-BiGRU is 16.57% under smoothing label processing, which is higher than that of ResNet. This implies that BiGRU can better excavate the dependence between the sleep stages with more comprehensive preservation of the internal sleep characteristics of the EEG signals. The overall recognition rate of MAResnet is 76.26%, i.e., more EEG sleep information can be learned compared to the traditional ResNets on a single scale. Also, the residual attention module is more accurate in extracting important information. The overall recognition rate of CNN-BiGRU is 7.78%, which is lower than that of the MAResnet-BiGRU, further illustrating the importance of the MAResnets in extracting important information. Finally, the overall recognition rate of MAResnet-BiGRU on Sleep-EDF dataset is 84.24%, and there is an increase in recall rates for each sleep stage. Among them, the recall for the S3 stage even reaches 92.28%.

Table 2. Recall rate of each sleep stage classification under different algorithms under label smoothing on Sleep-EDF dataset.

| Net model | W | S1 | S2 | S3 | REM | Total recognition rate |
|---|---|---|---|---|---|---|
| MAResnet-BiGRU | 88.24% | 67.2% | 83.53% | 92.28% | 88.95% | 84.24% |
| Resnet-BiGRU | 83.37% | 59.48% | 79.89% | 88.34% | 82.77% | 78.77% |
| CNN-BiGRU | 80.12% | 58.27% | 77.45% | 85.12% | 81.34% | 76.46% |
| MAResnet | 80.34% | 59.34% | 76.56% | 86.67% | 78.39% | 76.26% |
| CNN-GRU | 76.20% | 53.88% | 71.30% | 79.04% | 77.12% | 74.70% |
| ResNet | 69.20% | 49.32% | 60.26% | 67.38% | 64.84% | 62.20% |

**Table 3. The experimental results of Sleep-EDF data set are compared with the existing research results.**

| Number of output channels | Method | Author | Accuracy |
|---|---|---|---|
| 1 | XGBoost | Guo Yanping | 79.7% |
| 1 | K mean | Yu Ying et al | 72% |
| 1 | CNNs | Phan H et al | 82.6% |
| 1 | Residual CNNs | Humayun et al | 79.2% |
| 1 | Residual CNNs+BiLSTM | Seo H et al | 80.6% |
| 1 | MAResnet-BiGRU | Methods of this paper | 84.24% |

Compared to the above-mentioned network models, the classification recognition rate of MAResnet-BiGRU is improved by using multiple convolution kernels of different sizes in parallel at the same spatial position. Then, we obtain the output of the multiscale EEG sleep features and perform the multi-scale EEG sleep feature extraction by using the improved residual attention module in the neural network at each scale. Furthermore, the CAU and SAU units are added to enhance the EEG sleep characteristics associated with sleep classification by weakening the nonessential EEG sleep features. Finally, using BiGRU to model and classify the temporal feature information, more fully preserving the intrinsic sleep characteristics of EEG, to improve the sleep cycle classification recognition rate. Compared with the original Resnet network, the sleep staging recognition rate of MAResnet-BiGRU is increased by 22.04%. Hence, the results reveal the effectiveness of the proposed MAResnet-BiGRU network model.

Based on the Sleep-EDF data set. Table 3 compares the state-of-the-art research with the method proposed in this paper in terms of accuracy. In [35], single-channel EEG signals were used to perform five classifications by the XGBoost, where the classification accuracy reached 79.7%. [36] improved the K-means clustering algorithm, and the classification accuracy reached 72%. However, the above methods cannot achieve automatic feature extraction, and the classification accuracy is low. Phan H et al. [37] proposed a method to discriminatively learn a frequency-domain filter bank with a deep neural network (DNN) to preprocess the time-frequency image features. Humayun A I et al. [38] used the automatic deep learning method for PPG signals to evaluate sleep stages. The model has weak learning ability for the relationship between sleep debris, and the accuracy is only 79.2%. Seo H et al. [39] applied the convolution and long-short memory layer model to EEG sleep staging. The model lacks the acquisition of important information in the sequence, and the accuracy is 3.64% lower than that of the model in this paper.

Besides the sleep EDF data set, we also apply the model proposed in this paper to UCD data set and SHHS data set. On the UCD dataset, it is can be seen from Table 4, under the label smoothing treatment, the sleep staging recognition rate of the single-scale traditional ResNet model was lowest, only 59.70%, where the S1 stage recognition rate was even lower, only 45.32%. The overall recognition rates for models CNN-GRU, MAResnet, CNN-BiGRU and Resnet-BiGRU were 67.39%, 73.71%, 73.60% and 75.75%, respectively. The overall recognition rate of MAResnet-BiGRU on UCD dataset is 79.34%, and are higher than other models. In addition, the UCD data set is collected from patients with sleep diseases, so the accuracy of sleep classification is lower than that of sleep-EDF data set, which reflects the difficulty of sleep staging for patients with sleep diseases.

This model is also used in the SHHS data set. Table 5 compares the state-of-the-art research with the method put forward in this paper in terms of accuracy. As can be seen from Table 5. the overall recognition rate with this method on SHHS data set is 81.6%, higher than that of other models. Among all the comparison models, the classification accuracy of ResNet model

**Table 4. Recall rate of each sleep stage classification under different algorithms under label smoothing on UCD dataset.**

| Net model | W | S1 | S2 | S3 | REM | Total recognition rate |
|---|---|---|---|---|---|---|
| MAResnet-BiGRU | 84.50% | 63.10% | 79.50% | 86.30% | 83.30% | 79.34% |
| Resnet-BiGRU | 80.28% | 57.31% | 75.84% | 83.74% | 81.60% | 75.75% |
| CNN-BiGRU | 78.20% | 55.76% | 74.34% | 80.47% | 79.26% | 73.60% |
| MAResnet | 77.83% | 56.34% | 74.56% | 81.28% | 78.54% | 73.71% |
| CNN-GRU | 72.24% | 48.69% | 69.30% | 73.26% | 71.38% | 67.39% |
| ResNet | 64.37% | 43.10% | 61.20% | 66.31% | 63.52% | 59.70% |

is the lowest, only 60.24%. which is 21.26% lower than that of this paper. What's more, the classification accuracy of CNN-GRU, MAResnet, CNN-BiGRU and Resnet-BiGRU are 68.75%, 74.20%, 74.27% and 77.49%, respectively. Consequently it is not hard to draw a conclusion that the classification accuracy of other models are not as good as the one proposed in this paper.

In classification problems, the most common evaluation index is accuracy, which can directly reflect the correct proportion of points. But the actual classification problem, the number of samples of each category is often not very balanced. If the model is not adjusted on this imbalanced dataset, it is easy to give up small classes instead of large classes. The overall accuracy is high at this point, but some categories cannot be recalled at all. Kappa coefficient is used to evaluate the consistency between the classification model and the expert score [40]. Landis et al. think that the classification model with Kappa value greater than 0.80 is almost perfect, and the classification model with Kappa value between 0.61 and 0.80 has practical value [41]. To valuate the performance of the MAResnet-BiGRU model in this paper, we measure the classification model by calculating the kappa coefficient of the model, and the kappa coefficient is generally used as the evaluation of the multi-classification model. Specific calculations such as Eq (13):

$$k = \frac{p_0 - p_e}{1 - p_e} \qquad (13)$$

Among them, p0 is the sum of the number of samples correctly classified in each category divided by the total number of samples, that is, the overall classification accuracy, $p_e$e is the probability that the expected results are consistent with the real results, and the results are shown in Table 6. Table 6 provides cross validation results of Sleep-EDF dataset.

It can be seen from the table that the performance of the ResNet model is the weakest, and the kappa coefficient of the network model built in this paper is the highest, which indicates that the performance is optimal, and also reflects the high consistency between the automatic and manual stages of the model. It has good practical value.

**Table 5. Recall rate of each sleep stage classification under different algorithms under label smoothing on SHHS dataset.**

| Net model | W | S1 | S2 | S3 | REM | Total recognition rate |
|---|---|---|---|---|---|---|
| MAResnet-BiGRU | 85.24% | 64.37% | 82.90% | 89.29% | 86.2% | 81.60% |
| Resnet-BiGRU | 83.36% | 56.48% | 78.41% | 87.49% | 81.73% | 77.49% |
| CNN-BiGRU | 79.12% | 55.21% | 74.62% | 83.12% | 79.32% | 74.27% |
| MAResnet | 78.64% | 56.14% | 75.16% | 84.76% | 76.30% | 74.20% |
| CNN-GRU | 74.22% | 50.26% | 69.83% | 79.04% | 70.42% | 68.75% |
| ResNet | 65.10% | 45.32% | 63.74% | 65.28% | 61.8% | 60.24% |

**Table 6. Kappa coefficient under different algorithms.**

| Method | kappa coefficient |
|---|---|
| MAResnet-BiGRU | 0.79±0.03 |
| ResNet-BiGRU | 0.68±0.05 |
| CNN-BiGRU | 0.68±0.04 |
| MAResnet | 0.68±0.06 |
| ResNet | 0.57±0.08 |
| XGBoost | 0.70±0.06 |
| K mean | 0.63±0.06 |

As shown in Fig 7, the experimental results of Sleep-EDF data set are represented in a confusion matrix. The rows and columns of the confusion matrix represent the number of epochs for each sleep stage divided by sleep experts and our method, respectively, as shown in Fig 7. This is the statistical result of a random sample of 4182 samples for prediction. The diagonal is the number of correctly classified samples, and the other locations are the number of misclassified samples. It can be seen from Fig 7 that the classification effect of S1 stage is the worst. Among them, S1 stage is often confused with W, S3 and REM stages, which is consistent with the similarity of characteristic waves in each stage. The poor classification of the S1 phase may be due to the fact that the S1 phase, as the shortest transition phase, is most vulnerable to the combined effects of multiple, adjacent and similar classes. By observing the confusion matrix, it can be seen that the prediction results do not tend to occupy the S2 stage of most data, which shows that the method proposed in this paper alleviates the problem caused by category imbalance to a certain extent.

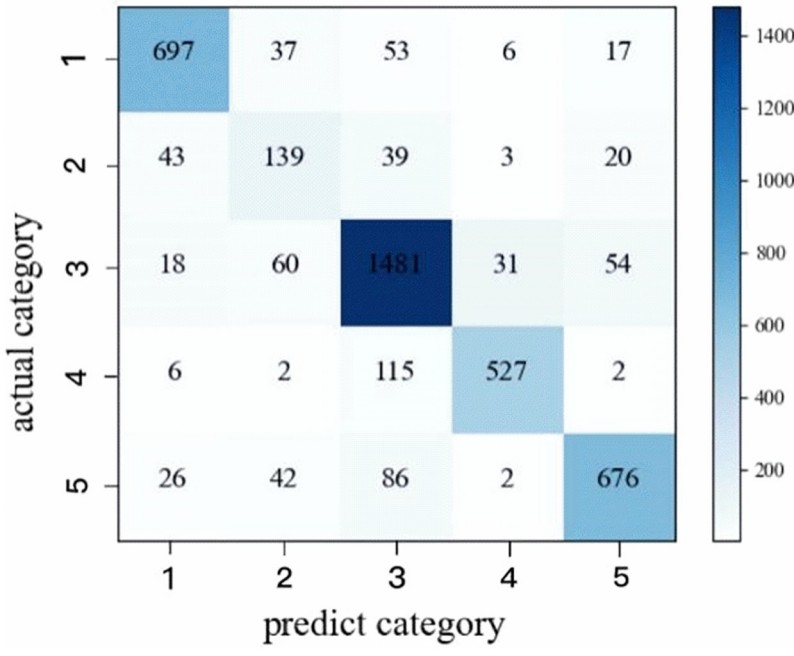

**Fig 7. Confusion matrix of Sleep-EDF data set.**

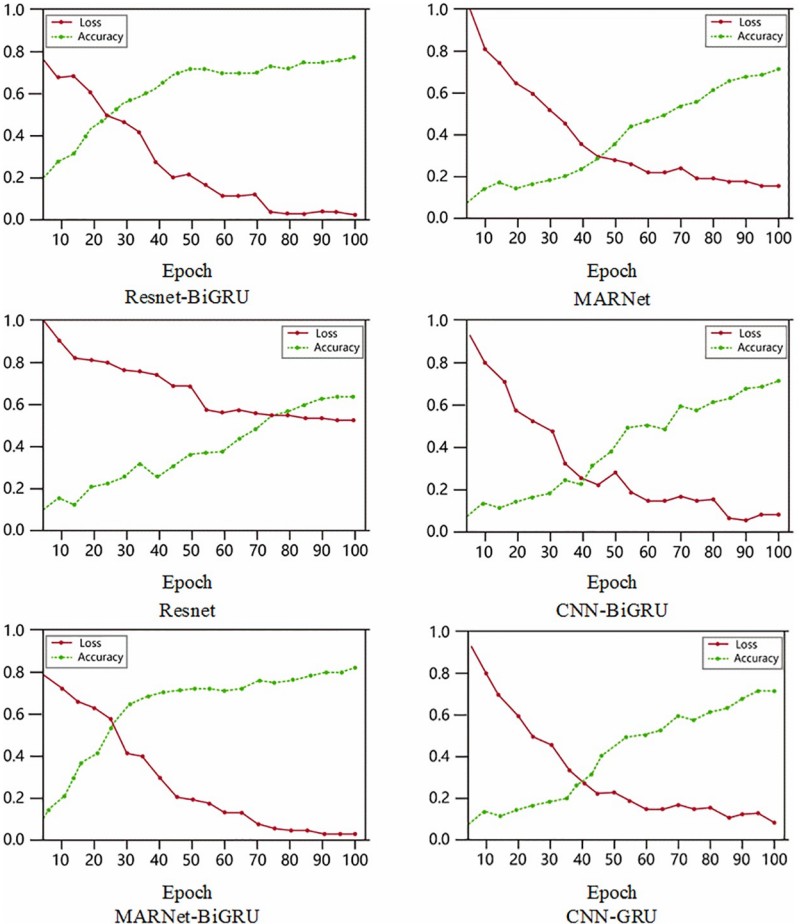

**Fig 8. On the Sleep-EDF dataset, training accuracy and loss value under different algorithms.**

In order to verify the classification effect of the improved network model and apply the model to sleep-EDF dataset, this model has been added several indicators such as loss function, recall rate and classification recognition rate of overall sleep cycle to evaluate the classification results. Under the same parameters, the MAResnet-BiGRU model is compared with the other five network models. The other five network models are Resnet-BiGRU, Resnet, CNN-BiGRU, MARNet and CNN-GRU. These six network models are classified and trained in the open source framework Pytorch. A total of 100 epochs are trained. The changes of accuracy and loss value in the training process are shown in Fig 8.

The green curve represents the change of accuracy with epoch, while the red curve represents the change of loss value with epoch. It can be seen from the figure that the accuracy increases and tends to be stable during training, and the loss value decreases and also tends to be stable, which shows that the six models in this paper have good staging performance for sleep EEG signals. In order to further compare the performance of the six models, the loss value curve is shown in one figure, as shown in Fig 9.

It can be seen from the figure that when the epoch is about 80, the curve of loss value tends to be flat, and it can be seen that the loss value of MAResnet-BiGRU model is the smallest, which proves that the prediction effect of the improved model offered by this paper is the best.

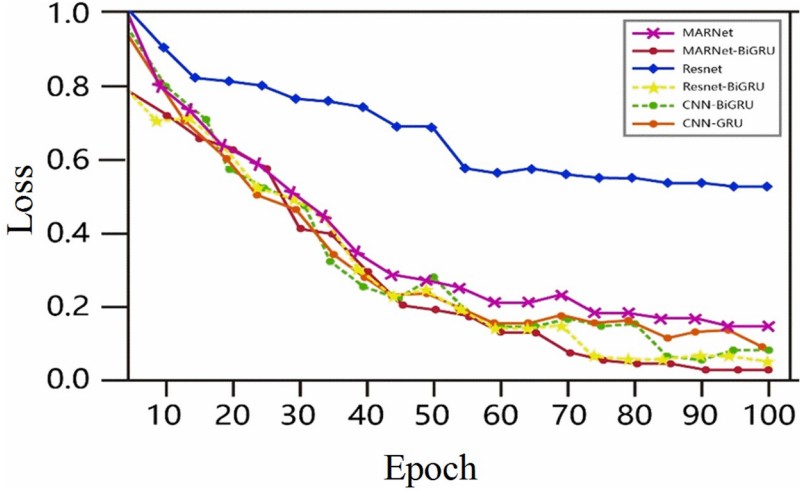

**Fig 9. On the Sleep-EDF dataset, training loss values of six models.**

In order to fully show the generalization performance of the model, the six models are 10-fold cross-validation with under label smoothing to obtain the overall recognition rate and recall rate in each period, as shown in Fig 10.

With the smooth label processing, the recognition rate of sleep stages of single-scale traditional Resnet network model is only 62.20%, and the recognition rate of S1 stage is only 49.32%. Compared with Resnet network model, the classification and recognition rate of sleep cycle of Resnet-BiGRU network model with the smooth label processing is 16.57% higher. The classification and recognition rate of CNN-BiGRU is 1.8% higher than that of CNN-GRU network model, which shows that BiGRU can better reflect the dependence of each sleep stage, and keep the intrinsic sleep characteristics of EEG signals more comprehensively. The overall recognition rate of MARNet model is 76.26%, which is 14.06% higher than that of the original Resnet network, which reflects that the improved algorithm architecture can learn more

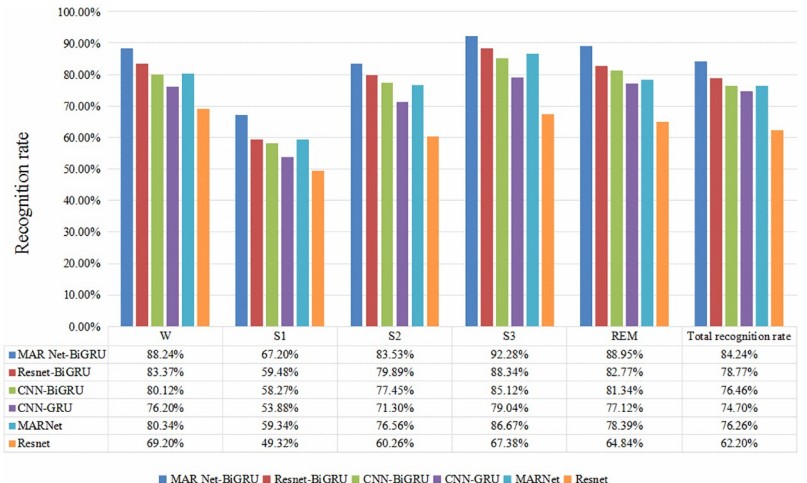

| | W | S1 | S2 | S3 | REM | Total recognition rate |
|---|---|---|---|---|---|---|
| MAR Net-BiGRU | 88.24% | 67.20% | 83.53% | 92.28% | 88.95% | 84.24% |
| Resnet-BiGRU | 83.37% | 59.48% | 79.89% | 88.34% | 82.77% | 78.77% |
| CNN-BiGRU | 80.12% | 58.27% | 77.45% | 85.12% | 81.34% | 76.46% |
| CNN-GRU | 76.20% | 53.88% | 71.30% | 79.04% | 77.12% | 74.70% |
| MARNet | 80.34% | 59.34% | 76.56% | 86.67% | 78.39% | 76.26% |
| Resnet | 69.20% | 49.32% | 60.26% | 67.38% | 64.84% | 62.20% |

**Fig 10. On the Sleep-EDF dataset, classification recall rates of each sleep period under different algorithms.**

information about EEG sleep, and the residual attention module constructed can extract important information more accurately. Compared with MARNet-BiGRU network model, the overall recognition rate of CNN-BiGRU network model is 7.78% lower, which further illustrates the important role of MARNet network in extracting key information. This paper constructed the MARNet-BiGRU network model' overall recognition rate is 84.24%, and its recall rate of each sleep stage is improved, among which the recall rate of S3 period is even 92.28%. Compared with the above five models, the classification recognition rate of the improved model is promoted to a certain extent. The recognition rate of sleep stages of the improved model is increased by 22.04% compared with that of the basic Resnet network before improvement. Experimental results prove the effectiveness of MARNet-BiGRU network model proposed in this paper.

## 5 Conclusions

Despite being useful, the existing sleep staging methods have several disadvantages, such as relying on artificial feature extraction, failing to recognize temporal sequence patterns in the long-term associated data, and reaching the accuracy upper limit of sleep staging. Hence, this paper proposes an automatic EEG sleep signal staging model, which integrates multi-scale ResNet and BIGRU through an attention mechanism. The proposed model is based on the residual neural network in deep learning. It uses convolution kernels of different sizes by adding the improved channel and spatial feature attention units to the traditional residual learning module in parallel at the same spatial position. Thus, multiscale feature extraction of the EEG sleep signals and residual learning of the neural networks is performed to avoid network degradation. Finally, BiGRU is used to determine the dependence between the sleep stages and to realize the automatic learning of sleep data staging features and sleep cycle extraction. The experimental results show that the classification accuracy and kappa coefficient of the MAResnet-BiGRU model are 84.24% and 0.7855, respectively on the Sleep-EDF data set. Compared with the traditional ResNet, the classification accuracy and kappa coefficient are improved by 22.04% and 0.2135, respectively. The classification accuracy of the proposed method on UCD data set and SHHS data set is 79.34% and 81.6% which is undoubtedly higher than other models. Thus effectiveness of the proposed method is proved.

## Supporting information

**S1 Data.**
(ZIP)

## Author Contributions

**Formal analysis:** Changyuan Liu, Okan K. Ersoy.

**Methodology:** Changyuan Liu, Okan K. Ersoy.

**Resources:** Changyuan Liu.

**Software:** Yunfu Yin, Yuhan Sun.

**Supervision:** Okan K. Ersoy.

**Writing – original draft:** Yunfu Yin.

**Writing – review & editing:** Yuhan Sun.

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
