## [Decision Letter · Decision Letter 0]

20 Apr 2022

PONE-D-22-09500Multi-scale ResNet and BiGRU automatic sleep staging based on attention mechanismPLOS ONE

Dear Dr. LIU,

Thank you for submitting your manuscript to PLOS ONE. After careful consideration, we feel that it has merit but does not fully meet PLOS ONE’s publication criteria as it currently stands. Therefore, we invite you to submit a revised version of the manuscript that addresses the points raised during the review process.

We look forward to receiving your revised manuscript.

Kind regards,

Sathishkumar V E

Academic Editor

PLOS ONE

Journal Requirements:

"Unfunded research"

"The authors would like to acknowledge the financial support received from The National Natural Science Foundation of China Number (51779050) and The National Natural Science Foundation of Heilongjiang Province Number(F2016022)."

"Unfunded research"

Reviewers' comments:

Reviewer's Responses to Questions

**Comments to the Author**

1. Is the manuscript technically sound, and do the data support the conclusions?

Reviewer #1: Partly

Reviewer #2: Yes

2. Has the statistical analysis been performed appropriately and rigorously? 

Reviewer #1: Yes

Reviewer #2: Yes

3. Have the authors made all data underlying the findings in their manuscript fully available?

Reviewer #1: No

Reviewer #2: Yes

4. Is the manuscript presented in an intelligible fashion and written in standard English?

Reviewer #1: Yes

Reviewer #2: Yes

5. Review Comments to the Author

Reviewer #1: this paper proposes an automatic EEG sleep signal staging model, which integrates multi-scale ResNet and BIGRU through an attention mechanism. The proposed model is based on the residual neural network in deep learning. It uses convolution kernels of different sizes by adding the improved channel and spatial feature attention units to the traditional residual learning module in parallel at the same spatial position. This paper is novel and the contributions are good for a journal paper. However, the following corrections to be made before consider this paper for publication in this journal.

1. The literature of the paper is poor, Recommended to provide the suitable literature by considering the recently published papers.

2. Some of the formulae of paper are available in the existing published papers.

3. There are several performance metrics in the literature, but the authors do not consider more. It is recommended to perform the experiments using multiple datasets and metrcs.

4. The reasons for achieving the superior performance to be listed. also list the limitations.

Reviewer #2: Abstract could be little more precise and crisp.

What audience would benefit most from this work?

What do you see as this manuscript’s main contribution to the literature in the Automated Sleep staging? Summarize in a few sentences.

What are the three strongest aspects of this manuscript?

However, I would like to encourage authors to include some more discussions and reviews of latest research works and existing techniques.

Any hyperparameter selection method employed?

In the introduction of the article, the study innovation should be clearly specified.

What are the three strongest aspects of this manuscript?

6. PLOS authors have the option to publish the peer review history of their article (what does this mean?). If published, this will include your full peer review and any attached files.

Reviewer #1: No

Reviewer #2: **Yes: **Usha Moorthy

---

## [Author Response · Author response to Decision Letter 0]

19 May 2022

Revised in accordance with editor and reviewe comments.

---

## [Decision Letter · Decision Letter 1]

23 May 2022

Multi-scale ResNet and BiGRU automatic sleep staging based on attention mechanism

PONE-D-22-09500R1

Dear Dr. LIU,

We’re pleased to inform you that your manuscript has been judged scientifically suitable for publication and will be formally accepted for publication once it meets all outstanding technical requirements.

Kind regards,

Sathishkumar V E

Academic Editor

PLOS ONE

Additional Editor Comments (optional):

Reviewers' comments:

Reviewer's Responses to Questions

**Comments to the Author**

1. If the authors have adequately addressed your comments raised in a previous round of review and you feel that this manuscript is now acceptable for publication, you may indicate that here to bypass the “Comments to the Author” section, enter your conflict of interest statement in the “Confidential to Editor” section, and submit your "Accept" recommendation.

Reviewer #1: All comments have been addressed

Reviewer #2: (No Response)

2. Is the manuscript technically sound, and do the data support the conclusions?

Reviewer #1: Yes

Reviewer #2: (No Response)

3. Has the statistical analysis been performed appropriately and rigorously? 

Reviewer #1: Yes

Reviewer #2: (No Response)

4. Have the authors made all data underlying the findings in their manuscript fully available?

Reviewer #1: Yes

Reviewer #2: (No Response)

5. Is the manuscript presented in an intelligible fashion and written in standard English?

Reviewer #1: Yes

Reviewer #2: (No Response)

6. Review Comments to the Author

Reviewer #1: The authors addressed all the comments and this version is recommended for publication. Congratulations to the authors..

Reviewer #2: (No Response)

7. PLOS authors have the option to publish the peer review history of their article (what does this mean?). If published, this will include your full peer review and any attached files.

Reviewer #1: No

Reviewer #2: **Yes: **Usha Moorthy

---

## [Editor Report · Acceptance letter]

27 May 2022

PONE-D-22-09500R1 

Multi-scale ResNet and BiGRU automatic sleep staging based on attention mechanism 

Dear Dr. LIU:

I'm pleased to inform you that your manuscript has been deemed suitable for publication in PLOS ONE. Congratulations! Your manuscript is now with our production department. 

Kind regards, 

on behalf of

Dr. Sathishkumar V E 

Academic Editor

PLOS ONE